# Design and Implementation of Multiband Noncontact Temperature-Measuring Microwave Radiometer

**DOI:** 10.3390/mi12101202

**Published:** 2021-09-30

**Authors:** Guangmin Sun, Jie Liu, Jingyan Ma, Kai Zhang, Zhenlin Sun, Qiang Wu, Hao Wang, Yiming Liu

**Affiliations:** 1Faculty of Information Technology, Beijing University of Technology, Beijing 100124, China; gmsun@bjut.edu.cn (G.S.); KaiZhang@emails.bjut.edu.cn (K.Z.); S201961721@emails.bjut.edu.cn (Z.S.); wuqiang@bjut.edu.cn (Q.W.); wanghao@bjut.edu.cn (H.W.); 2School of Electrical Engineering, North China Electric Power University, Beijing 100096, China; 120192301623@ncepu.edu.cn

**Keywords:** multiband, novel interferometric microwave radiometer, multilayer tissues, high sensitivity

## Abstract

In this paper, a multiband noncontact temperature-measuring microwave radiometer system is developed. The system can passively receive the microwave signal of the core temperature field of the human body without removing the clothes of the measured person. In order to accurately measure the actual temperature of multilayer tissue in human core temperature field, four frequency bands of 4–6 GHz, 8–12 GHz, 12–16 GHz, and 14–18 GHz were selected for multifrequency design according to the internal tissue depth model of human body and the relationship between skin depth and electromagnetic frequency. Used to measure the actual temperature of human epidermis, dermis, and subcutaneous tissue, a small and highly directional multiband angular horn antenna was designed for the radiometer front end. After the error analysis of the full-power microwave radiometer, a novel hardware architecture of the microwave interferometric temperature-measuring radiometer is proposed, and it is proven that the novel interferometric microwave radiometer has less error uncertainty through theoretical deduction. The experimental results show that the maximum detection sensitivity of the novel interferometric microwave temperature-measuring radiometer is 215 mV/dBm, and the temperature sensitivity is 0.047 K/mV. Compared with the scheme of the full-power radiometer, the detection sensitivity is increased 7.45-fold, and the temperature sensitivity is increased 13.89-fold.

## 1. Introduction

At present, temperature measurement can be divided into contact temperature measurement and noncontact temperature measurement. As a representative of contact temperature measurement, the mercury thermometer has the advantages of high accuracy and stable performance, but its measurement speed is slow and cross-infection is possible [1]. Noncontact temperature measurement technology, represented by infrared measurement, mainly monitors the temperature of the human skin. However, there is a certain difference between the internal tissue temperature of the human body and the skin surface temperature [2,3]. For healthy people, the difference between the temperature of core organs such as the heart and skin temperature can reach ±2 °C throughout the circadian cycle [4]. The temperature changes of tissues and organs at different depths can reflect the health of the lesion. It is very necessary to develop radiometers that monitor the internal tissue temperature of the human body in clinical applications.

Because clothing has a very weak attenuation effect on microwave signals, a radiometer in the microwave band can passively receive the microwave signal of human body temperature radiation through clothing, which allows measuring the true temperature of the human body. Microwave radiometers can also provide real-time, noncontact temperature monitoring for intensive care unit patients at appropriate distances, reducing the number of times healthcare workers come into contact with patients. Especially in high-risk infectious wards, the probability of infection can be greatly reduced [5]. In addition, the application of noncontact temperature measuring equipment can speed up the emergency triage process and buy more treatment time.

A microwave radiometer can passively receive the electromagnetic wave radiated by the human body [6]. After the received signal is analyzed and processed, the information representing the temperature of the human body can be obtained. At present, researchers have made a series of achievements in the field of noncontact measurement of human body temperature [7,8] and human subcutaneous tissue temperature using microwave radiometers [9,10,11,12].

In 2015, Pi et al. used a C-band receiver to match a 4 GHz monopole antenna for contact and noncontact measurement of water temperature. In contact measurement, the temperature sensitivity was 0.051 K/mV [13]. In the noncontact measurement, the temperature sensitivity of the C-band radiometer was 0.11 K/mV by matching the temperature of the load and the output voltage. In 2016, Hao et al. designed a full-power dual-polarization radiometer with a center frequency of 89 GHz [14]. The bandwidth of the designed radiometer was 2 GHz, the detection sensitivity of the radiometer was 28.75 mV/dBm, and the temperature sensitivity of the radiometer was about 2.3 K/mV. In 2020, Liu et al. designed a full-power radiometer structure for temperature measurement in the full W-band [15]. A grid voltage compensation circuit was designed for the low-noise amplifier in the receiver to offset the drift of the detection diode to a certain extent. These researchers have laid a great foundation for the study of temperature measurement using radiometers. Radiometers are mainly designed on the basis of a full-power microwave radiometer architecture for different scenarios. However, the temperature sensitivity of radiometers still cannot meet the medical standard [16,17]. The bandwidth coverage of a microwave radiometer is small, which is not enough to measure the temperature of the core temperature field of the human body. Therefore, this paper proposes a multiband noncontact temperature measurement microwave radiometer system based on a novel interferometric microwave radiometer architecture. The goal was to achieve a wide-band, high-sensitivity microwave temperature radiometer for the noncontact measurement of human body temperature and subcutaneous tissue temperature.

From 2000 to 2004, Hand et al. designed a five-band Dicke radiometer system to monitor the temperature of infants’ central brain [18,19,20]. A body model of neonatal brain temperature distribution was established. In 2015, He Fan, on the basis of the relative position and electromagnetic characteristics of human tissues, established a human body model for temperature measurement of the human heart, which included five layers: skin, fat, muscle, bone, and heart [21]. However, the above studies mainly focused on the temperature monitoring of specific organs of the human body. In this paper, as a function of the relationship between skin depth and microwave frequency, and considering the human tissue depth model, the temperature of tissues and organs in different parts of the human body is measured by the corresponding antenna.

In 2013, Wang Bin designed a Ka-band AC radiometer to measure human skin temperature by matching a direct detection receiver with a Cassegrain antenna [22]. In 2019, Zhang et al. adopted a better directional pyramidal horn antenna for the front end of a full-power radiometer [23]. In order to measure the temperature of the core temperature field inside the human body in the lower frequency band, this paper describes the design of a small highly directional multifrequency cone horn antenna for the temperature measurement of human tissue.

In 2018, Ubaichin et al. adopted an S-band full-power microwave radiometer [24]. Due to the nonlinear characteristics of the square law detector, the low-frequency component of the output of the detector also contained other frequency components, thus leading to nonlinear distortion of the radiometer. In order to further explore the reason for the error of the full-power microwave radiometer, this paper discusses the relationship model between component error and temperature uncertainty in the full-power microwave radiometer system. A novel type of interferometric microwave radiometer is proposed, and the uncertainty of the temperature measurement error of the novel interferometric microwave radiometer is proven to be smaller by simulation verification and error analysis. Furthermore, the mathematical model of temperature uncertainty of the novel interferometric microwave radiometer is analyzed. A prototype of the novel interferometric microwave radiometer is constructed, and the performance index of the novel interferometric microwave radiometer is compared with the full-power microwave radiometer.

The technical route of the overall scheme proposed in this paper is shown in Figure 1. Firstly, the RF front-end of the system is designed. According to the relationship between skin depth and microwave frequency, small and highly directional horn antennas of the corresponding frequency bands are designed. Then a full-power microwave radiometer is built, and a novel interferometric microwave radiometer with higher sensitivity is proposed and built. The uncertainty of the temperature measurement error of the two radiometers is analyzed, and the performance of the complex correlator is simulated. Lastly, the performance of the system is verified by comparing the sensitivity of temperature measurement and detection of two kinds of temperature radiometers.

## 2. Relationship between Multilayer Tissue Measurement and Microwave Frequency Band

Figure 2 simply illustrates a model of human tissue depth. The human epidermis is 0.2–1.5 mm thick and consists of a laminated flat epithelium, which usually reaches its maximum thickness on the palmate surface of the limbs. Below is the dermis, which is about 0.3–3 mm thick and consists mainly of various proteins. Further down is the subcutaneous tissue, with a thickness of about 3–10 mm, which is mainly composed of fat, but also some connective tissue; it is rich in capillaries, and the abdomen, buttocks, and thighs are usually the thickest.

Considering that the human body is made up of many tissues, each with a different medium, its dielectric constant, conductivity, and oxygen content in red blood cells will affect electromagnetic signals during noncontact human body temperature measurement [25]. The penetration of electromagnetic waves in a standard measurement regime is defined as the skin depth. The skin depth of dry skin and wet skin is mainly determined by their dielectric coefficient and electrical conductivity, and a change in the corresponding relationship is mainly caused by a change in blood glucose level. In a certain time range, it can be assumed that the corresponding relationship between the skin depth and frequency is consistent [26].

Figure 3 shows the relationship between skin depth and electromagnetic frequency on dry skin and wet skin. By analyzing the corresponding relationship between skin depth and the human tissue model, we chose four frequency bands of 4–6 GHz, 8–12 GHz, 12–16 GHz, and 14–18 GHz to design a multifrequency noncontact temperature radiometer for temperature measurement of the human skin layer, dermis, subcutaneous tissue. Table 1 shows the corresponding relationship between the frequency bands of the multiband noncontact thermometer radiometer system and skin depth and human tissue.

## 3. Hardware Circuit Design of Multiband Microwave Temperature-Measuring Radiometer

### 3.1. Design of Multi-Band Pyramidal Horn Antenna

In order to detect the temperature of the three layers of human skin tissue, and considering the noncontact method of human body temperature detection, this paper designed a pyramidal horn antenna for the front end of the microwave temperature-measuring radiometer. In this paper, the angular horn antenna was connected to the rectangular waveguide, and the open surface of the waveguide was gradually enlarged. By changing the matching relationship between the waveguide and the free space, the reflection coefficient of energy was reduced. Thus, the microwave signal of human body temperature radiation could be better received.

The angular cone horn antenna is formed by the gradual expansion of the feed waveguide at a certain angle. Antenna schematic diagram of E-plane and H-plane are shown in Figure 4. The gradual expansion of the waveguide interface improves the match between the waveguide and the free space, making the reflection coefficient smaller [26]. If the aperture field of the horn antenna is uniformly distributed, then the directivity of horn antenna can be expressed as [27]: (1)$$D_{dB}=10log\left[\frac{\pi }{8}\cdot \frac{D_{E}D_{H}}{\lambda^{2}}\cdot \frac{X\left(s\right)}{s}\cdot \frac{Y\left(t\right)}{t}\right]$$
where $D_{E}$ and $D_{H}$ are the length of the antenna aperture on the E-plane and H-plane respectively. a and b are the length of the antenna aperture on the E-plane and H-plane respectively. X(s) and Y(t) are parameters related to the quadratic phase distribution constants, respectively. The quadratic phase distribution constants s and t can be expressed as:(2)s=18(Aλ)21R1λ=DE(DE−a)8λR
(3)t=18(DHλ)21R2λ=DH(DH−b)8λR

The optimum unconstrained directivity occurs when t=0.375 and s=0.25. In order to reduce the interference of the radiation signal received from other objects during the measurement of human body temperature. It is necessary to design an antenna with the smallest aperture possible, so that the antenna receives radiation signals only to the human body. Therefore, this paper considers the design of small aperture antenna by adjusting the E-plane beam angle $HP_{E}$ (deg) and H-plane beam angle $HP_{H}$ (deg).

To design a small, highly directional antenna that can measure the temperature of the core tissues of the human body. In this paper, half-power beam width and antenna direction is taken as the constraint condition [28]. Therefore, it is necessary to know the relationship between the constants s and t and the corresponding half-power beamwidth. Then, through the fitting data, two approximate expressions are obtained, namely:(4)$$\frac{D_{E}}{\lambda }\frac{\tan\left(\frac{\theta_{3dB}}{2}\right)}{\cos\left(\frac{\theta_{3dB}}{2}\right)}=\frac{1}{2}\sqrt{\frac{a_{H}+c_{H}t^{2}+e_{H}t^{4}+g_{H}t^{6}+i_{H}t^{8}}{1+b_{H}t^{2}+d_{H}t^{4}+f_{H}t^{6}+h_{H}t^{8}}},\;t\leq 0.88$$
(5)$$\frac{D_{H}}{\lambda }\frac{\tan\left(\frac{\theta_{3dB}}{2}\right)}{\cos\left(\frac{\theta_{3dB}}{2}\right)}=\frac{1}{2}\sqrt{\frac{a_{E}+c_{E}s^{2}+e_{E}s^{4}+g_{E}s^{6}}{1+b_{E}s^{2}+d_{E}s^{4}+f_{E}s^{6}+h_{E}s^{8}}},\;s\leq 0.47$$
where $a_{H}$, $b_{H}$, $c_{H}$, $d_{H}$, $e_{H}$, $f_{H}$, $g_{H}$, $h_{H}$, $i_{H}$, $a_{E}$, $b_{E}$, $c_{E}$, $d_{E}$, $e_{E}$, $f_{E}$, $g_{E}$ and $h_{E}$ are approximate coefficients.

In the process of antenna design, the beam angle $HP_{E}^{0}$ and $HP_{H}^{0}$ of horn antenna can be calculated for the given sizes of $HP_{E}$ and $HP_{H}$ without constraint. In order to design the optimum horn under constraints on the half-power beam width, we have found that one of the parameters, s or t, has to retain the value t=0.375 and s=0.25 of the optimum horn, while the other parameter changes. In this paper, the required ratio k of E-plane and H-plane beams is introduced to determine the adjustment mode of the two beam angles. And k is expressed as:(6)$$k=\frac{HP_{H}}{HP_{E}},\;k_{opt}^{E}=\frac{HP_{H}^{0}}{HP_{E}},\;k_{opt}^{H}=\frac{HP_{H}}{HP_{E}^{0}}$$

When $k\leq \min\left(k_{opt}^{E},\;k_{opt}^{H}\right)$, t must be equal to 0.375, and the constant s becomes:(7)$$s={\left(a_{0}+\frac{a_{1}}{k^{2}}+\frac{a_{2}}{D_{dB}}\right)}^{-1}$$

The directivity $D_{dB}$ is:(8)$$D_{dB}=c_{0}\left(k\right)+c_{1}\left(k\right)\ln\left(HP_{H}\right)$$
where $c_{0}\left(k\right)$ and $c_{1}\left(k\right)$ are functions of *k*, and the approximate coefficients of the functions are known.

When $k\geq \max\left(k_{opt}^{E},\;k_{opt}^{H}\right)$, s must be equal to 0.25, and the constant t becomes:(9)$$t=\frac{d_{0}+d_{1}k+d_{2}k^{2}+d_{3}D_{dB}+d_{4}{D_{dB}}^{2}+d_{5}{D_{dB}}^{3}}{1+d_{6}k+d_{7}D_{dB}+d_{8}{D_{dB}}^{2}}$$

The directivity $D_{dB}$ is:(10)$$D_{dB}=f_{0}\left(k\right)+f_{1}\left(k\right)\ln\left(HP_{E}\right)$$
where $f_{0}\left(k\right)$ and $f_{1}\left(k\right)$ are functions of k, and the approximate coefficients of the functions are known.

When $\min\left(k_{opt}^{E},\;k_{opt}^{H}\right)\leq k\leq \max\left(k_{opt}^{E},\;k_{opt}^{H}\right)$, the modification of s and t should be considered respectively to find the optimal solution by comparison.

According to the above three situations, $D_{E}$ and $D_{H}$ can be obtained by substituting the parameters s, t and the wavelength λ into Equations (4) and (5). Then, by substituting parameters s, t, $D_{E}$ and $D_{H}$ into Equations (2) and (3), the antenna depth R can be calculated. In this paper, the H-plane beam is sacrificed to optimize the E-plane beam, so that the designed angular horn antenna has a smaller antenna aperture area and a reasonable beam angle. Table 2 shows the size of the corresponding pyramid horn antenna designed.

On the basis of the antenna size calculated above, this paper used COMSOL Multiphysics software to carry out the simulation design. The simulation results of the antenna index measured in each frequency band are shown in Table 3.

In Table 3, pre/post suppression ratio refers to the ratio of the signal radiation intensity of the antenna in the direction of the main lobe and the direction of the rear lobe. It can be seen that the angular horn antenna designed in this paper had the characteristics of large far-field gain, small aperture area, and small 3 dB beam angle. 

Following design, a third-party organization was invited to make the antenna according to the designed 3D CAD drawings using machine tool cutting. The process is shown in Figure 5, and the cone horn antenna is shown in Figure 6. The final mechanical tolerance of the designed antenna was ±1%.

### 3.2. Circuit Design of Full-Power Microwave Temperature Measurement Radiometer

Previous studies on microwave radiometer temperature measurement mainly adopted a full-power structure. In the existing literature, a full-power radiometer with high sensitivity in a single band was mainly implemented. Therefore, a multiband full-power microwave temperature-measuring radiometer was first constructed in this paper. The architecture of the multiband full-power microwave temperature-measuring radiometer included a multiband angular horn antenna, low-noise amplifier (LNA), cavity filter, RMS power detector, and analog-to-digital converter (ADC). The block diagram and prototype are shown in Figure 7 and Figure 8, respectively.

In order to verify the feasibility of the full-power microwave radiometer, a complex correlator was added to the structure of the full-power microwave radiometer to obtain the output correlator circle [29]. The main function of the complex correlator is to extract useful information from interference and noise, compare the two input signals, and obtain their phase difference, so as to obtain useful information in the signal [30].

Because the full-power microwave radiometer uses an RMS power detector to realize the detection function, and because an analog complex correlator has higher theoretical accuracy than a digital complex correlator and is more suitable for circuits with wider system bandwidth, the full-power microwave temperature-measuring radiometer was designed to use an additive analog complex correlator [17]. A block diagram of the full-power microwave radiometer characteristics is shown in Figure 9.

In this paper, sinusoidal excitation was added to the front end of the complex correlator for frequency domain simulation, and the actual device parameters were introduced into the simulation model for simulation. The phase was swept at a 0°–360° phase difference and fixed frequency, and the points corresponding to the data were plotted and combined to form a complex correlation circle with the in-phase correlation component and orthogonal correlation component as coordinates. The shape and center position of the generated complex correlator can intuitively reflect whether the internal circuit of the complex correlator is distorted. The complex correlation circle generated by the ideal complex correlator has no deviation in its center and no distortion in its shape [30]. Figure 10 shows a block diagram of the simulation of the full-power microwave radiometer using ADS software.

In order to ensure the accuracy and authenticity of the simulation results, a vector network analyzer was used to measure the S-parameters of each module and connection line in the microwave radiometer during the verification process. The corresponding SNP file was added to the simulation system of the full-power microwave radiometer. 

In this paper, a phase scanning simulation was carried out when the input end of microwave radiometer was connected to a 50 Ω standard load. According to the formula of equivalent noise temperature, the corresponding temperature *T* under standard resistance can be solved. At the same time, the input power can be calculated by substituting the corresponding temperature *T* into Equation (13) of thermal noise.
(11)$$\Gamma_{REF}=\frac{Z_{0}-R_{REF}}{Z_{0}+R_{REF}},$$
(12)$$T_{REF}=\left(1-{\left|\Gamma_{REF}\right|}^{2}\right)T_{\gamma },$$
(13)P=kTB,
where $Z_{0}=50\;\Omega$, $T_{\gamma }=290\;K,$ and $R_{REF}=50\;\Omega$. The corresponding equivalent noise temperature can be calculated as 290 K using Boltzmann constant $k=1.38\times 10^{-23}\;J/K$ and operating bandwidth B=4 GHz in Equation (13). It can be calculated that the equivalent noise power is −78 dBm when a standard load is connected, and the complex correlation circle obtained by the corresponding phase scan is shown in Figure 11.

It can be seen from the simulation results that the shape of the generated correlator circle was changed, which means that the complex correlator constructed with a full-power structure was not ideal.

### 3.3. Influence of Different Types of Radiation Sources on Full-Power Microwave Radiometer

It can be seen from Figure 11 that the complex correlation circle generated by the phase sweep of the full-power structure was not ideal. In this paper, error analysis of the full-power structure was carried out to analyze the influence of the error of the simulated complex correlator on high-precision temperature measurement. Analog complex correlators can be divided into two types: additive and multiplicative. The composition of the additive analog complex correlator is shown in Figure 12.

An additive analog complex correlator is composed of power splitter, orthogonal coupler, diode square law detector, and differential amplifier. A 0° power splitter is used, and the phase of each power splitter on the output phase reference surface should be strictly equal. The phase of signal input terminal IN of the 3 dB orthogonal coupler is equal to that of the direct terminal 0, and the phase difference between the signal input terminal IN and the direct terminal 0 and coupling terminal −90 is 90°. Diode square law detectors are used to detect power. The differential amplifier is used to subtract the differential mode signal of two signals and to amplify the common mode signal.

Based on the principle in Figure 12, this paper deduced the relationship of the unbalance which affects the sensitivity and precision of the full-power microwave radiometer.

It can be obtained from Figure 12 that the input of the additive analog complex correlator can be expressed as
(14)$$A_{1}\left(t\right)=a_{1}\cos\left(2\pi ft\right),A_{2}\left(t\right)=a_{2}\cos\left(2\pi ft-\phi \right)m$$
where $a_{1}$ and $a_{2}$ are the amplitudes of the input signal. After the relevant operation of the signal through the power divider and coupler, and then through the square law detector and the low-pass filter, the output voltage of the analog complex correlator can be obtained from the differential amplifier (the gains of the differential amplifier are $G_{1}$ and $G_{2}$ respectively) as follows:(15)$$I_{1}=\frac{G_{1}Ka_{1}a_{2}}{2}\sin\phi ,\;I_{2}=\frac{G_{2}Ka_{1}a_{2}}{2}\cos\phi .$$

When the radiometer uses a wide-band thermal emitter input, the response is integrated over the entire bandwidth of the receiver. For the additive analog complex correlator, the output results of the additional amplitude error and phase error are expressed as
(16)$$V_{REAL}\left(\varphi \right)=\int_{f_{C}-\frac{∆f_{RF}}{2}}^{f_{C}+\frac{∆f_{RF}}{2}}E_{I}^{\prime }\left(f\right)df,V_{IMAG}\left(\varphi \right)=\int_{f_{C}-\frac{∆f_{RF}}{2}}^{f_{C}+\frac{∆f_{RF}}{2}}E_{I^{\prime }}^{\prime }\left(f\right)df.$$

The power splitter PWR1 and 3 dB orthogonal couplers HYB1, HYB2, and HYB3 in the complex correlator are assumed to be unbalanced in power and phase. By setting the phase difference of the output of two ports of the power splitter PWR1 as $\varphi_{1}\left(f\right)$, the output signals can be expressed as
(17)$$B_{1}=(k_{1}\left(f\right)/\sqrt{2})\cos\omega t,\;B_{2}=(k_{2}\left(f\right)/\sqrt{2})\cos\omega t$$
(18)$${\left(k_{1}\left(f\right)/\sqrt{2}\right)}^{2}+{\left(k_{2}\left(f\right)/\sqrt{2}\right)}^{2}=1$$.

The phase differences in the 3 dB orthogonal coupler HYB1 are $90{}^{\circ}+\varphi_{11}\left(f\right)$ and $90{}^{\circ}+\varphi_{12}\left(f\right)$. The amplitude imbalances are $n_{11}\left(f\right)$ and $n_{12}\left(f\right)$. The phase differences in the 3 dB orthogonal coupler HYB1 are $90{}^{\circ}+\varphi_{21}\left(f\right)$ and $90{}^{\circ}+\varphi_{22}\left(f\right)$. The amplitude imbalances are $n_{21}\left(f\right)$ and $n_{22}\left(f\right)$. The phase differences in the 3 dB orthogonal coupler HYB1 are $90{}^{\circ}+\varphi_{31}\left(f\right)$ and $90{}^{\circ}+\varphi_{32}\left(f\right)$. The amplitude imbalances are $n_{31}\left(f\right)$ and $n_{32}\left(f\right)$.

Furthermore, assuming that all parts of the microwave front end of the interferometric correlator are ideal except the analog complex correlator, then $a_{1}=a_{2}=a$. At the same frequency, all power splitters cause the same amplitude and phase unbalance, and all 90° couplers cause the same amplitude and phase unbalance. Thus,
(19)$$n_{11}\left(f\right)=n_{21}\left(f\right)=n_{31}\left(f\right)=n_{1}\left(f\right),\;n_{12}\left(f\right)=n_{22}\left(f\right)=n_{32}\left(f\right)=n_{2}\left(f\right),$$
(20)$$\;\varphi_{11}\left(f\right)=\varphi_{21}\left(f\right)=\varphi_{31}\left(f\right),\;\varphi_{12}\left(f\right)=\varphi_{22}\left(f\right)=\varphi_{32}\left(f\right),$$
(21)$$K_{1}\left(f\right)=K_{2}\left(f\right)=K_{R}\left(f\right),\;K_{3}\left(f\right)=K_{4}\left(f\right)=K_{I}\left(f\right).$$

Suppose that the DC component has been removed by certain means. Then,
(22)$$k_{1}\left(f\right)n_{1}^{2}\left(f\right)n_{2}\left(f\right)K_{R}\left(f\right)=k_{2}\left(f\right)n_{2}^{2}\left(f\right)n_{1}\left(f\right)K_{I}\left(f\right)=1,$$
(23)$$\emptyset \left(f\right)=-\varphi_{1}\left(f\right)+\varphi_{12}\left(f\right).$$

For the case where φ is a single constant value, if only the phase error exists in the real and imaginary parts of the analog complex correlator, then
(24)$$\begin{array}{c} V=\frac{a^{2}}{2}∆f_{RF}{\left[\cos^{2}\left(∆\varphi_{1}\right)\cos^{2}\left(\varphi +∆\varphi_{2}\right)+\cos^{2}\left(∆\varphi_{3}\right)\sin^{2}\left(\varphi +∆\varphi_{4}\right)\right]}^{\frac{1}{2}},\\ =\frac{a^{2}}{2}∆f_{RF}{\left[\cos^{2}\left(∆\varphi_{1}\right)\cos^{2}\left(\varphi_{t}\right)+\cos^{2}\left(∆\varphi_{3}\right)\sin^{2}\left(\varphi_{t}+∆\varphi \right)\right]}^{\frac{1}{2}}, \end{array}$$
where $\varphi_{t}=\varphi +∆\varphi_{2}$ and $∆\varphi_{\;}=∆\varphi_{4}-∆\varphi_{2}$. Temperature changes caused by system noise, gain fluctuation, and phase error are statistically independent. For the case where φ is a single value, the total root-mean-square uncertainty can be expressed as
(25)$$∆T\;=\;{\left[{\left(∆T_{N}\right)}^{2}+{\left(∆T_{G}\right)}^{2}+{\left(∆T_{\phi }\right)}^{2}\right]}^{\frac{1}{2}},$$
where
(26)$$∆T_{\phi }=T_{sys}\left[1-\left(\cos^{2}(∆\varphi_{1}\right)\cdot \cos^{2}(\varphi_{t}\right)+\cos^{2}(∆\varphi_{3})\sin^{2}(\varphi_{t}+∆\varphi ){)}^{\frac{1}{2}}].$$

### 3.4. Influence of Wide-Band Thermal Radiation Source on Interferometric Microwave Radiometer

Figure 13 is a block diagram of the multiplicative analog complex correlator applied to the interferometric microwave radiometer architecture [31]. It consists of a power splitter, 3 dB orthogonal coupler, mixer, and differential amplifier.

When the input of the multiplicative analog complex correlator is as shown in Equation (14), signals $F_{1}$, $F_{2}$, $H_{1}$, and $H_{2}$ are passed through a differential amplifier (the gains of the differential amplifier are $G_{1}$ and $G_{2}$, respectively) and an ideal low-pass filter, whereby output signals $I_{1}$ and $I_{2}$ are obtained, which are expressed as
(27)$$I_{1}=\frac{G_{1}a_{1}a_{2}}{2}\cos\left(\varphi \right),\;I_{2}=\frac{G_{2}a_{1}a_{2}}{2}\sin\left(\varphi \right)$$

The power splitter PWR1 and 3 dB orthogonal coupler HYB1 in the complex correlator are assumed to be unbalanced in power and phase. By setting the phase difference of output of two ports of power splitter PWR1 as $\varphi_{1}\left(f\right)$, the output signal can be expressed as
(28)$$\begin{array}{c} B_{1}=(k_{1}\left(f\right)/\sqrt{2})\cos\omega t,\;B_{2}=(k_{2}\left(f\right)/\sqrt{2})\cos\omega t,\\ {\left(k_{1}\left(f\right)/\sqrt{2}\right)}^{2}+{\left(k_{2}\left(f\right)/\sqrt{2}\right)}^{2}=1 \end{array}$$.

When the signal is transmitted to 3 dB orthogonal coupler HYB1, the phase difference between the coupling port and the through port is $90{}^{\circ}+\varphi_{11}\left(f\right)$. Moreover, it is assumed that all parts of the microwave front end of the interferometric correlator are ideal except the analog complex correlator. After transformation, the amplitude of the signal entering the analog complex correlator is equalized. Thus, a1 = a2 = a. At the same frequency, $k_{1}\left(f\right)=k_{2}\left(f\right)=k\left(f\right)$. Then, the output results of the additional amplitude error and phase error are expressed as
(29)$${I_{1}}^{\prime }\left(f\right)=\frac{1}{2}a^{2}H_{1}\left(f\right)\cos\left(\varphi \right),\;{I_{2}}^{\prime }\left(f\right)=\frac{1}{2}a^{2}H_{2}\left(f\right)\sin\left(\varphi_{1}\left(f\right)-\varphi -\varphi_{11}\left(f\right)\right)$$
where $H_{1}\left(f\right)=k\left(f\right)n_{11}\left(f\right)$, and $H_{2}\left(f\right)=k\left(f\right)n_{12}\left(f\right)$. To simplify the calculation, suppose that $H_{1}\left(f\right)=H_{2}\left(f\right)=H\left(f\right)=1$.

If only the phase error exists in the real and imaginary parts of the analog complex correlator, it can be obtained according to Equations (16) and (27).
(30)$$V=\sqrt{{\left(V_{REAL}\right)}^{2}+{\left(V_{IMAG}\right)}^{2}}=\frac{1}{2}a^{2}∆f_{RF}{\left(\cos^{2}\left(\varphi \right)+\sin^{2}\left(\varphi +∆\varphi_{1}\right)\right)}^{\frac{1}{2}},$$
where ∆φ = $\varphi_{11}\left(f\right)-\varphi_{1}\left(f\right)$. For the case where φ is a single value,$\;∆T_{\varphi }$ is expressed as
(31)$$∆T_{\varphi }=T_{sys}\left[\;1-{\left(\cos^{2}\left(\varphi \right)+\sin^{2}\left(\varphi +∆\varphi \right)\right)}^{\frac{1}{2}}\right].$$

Compared with Equations (26) and (31), the phase error of the multiplicative complex correlator depends only on the power splitter and a coupler. Compared with the additive complex correlator, the phase error is smaller, and the temperature uncertainty is also smaller.

### 3.5. Design of Novel Interferometric Microwave Temperature Radiometer

Considering the influence of the unbalance on the microwave radiometer, this paper selected an interferometric microwave radiometer based on multiplicative complex correlator for the design of the prototype, and a novel interferometric microwave radiometer is proposed. The measured signal enters the receiver through the antenna and is divided into two signals of the same amplitude through the power divider. One signal is down-converted by a mixer, while the other signal is down-converted by a mixer and then phase-shifted by a phase shifter. Then, the two signals are correlated in the analog complex correlator, and the DC signal in its output is the power value of the measured signal.

At the same time, the effect of a nonideal transmission line on the complex correlator system was considered. In this paper, a wideband orthogonal demodulator was used to replace the multiplicative analog complex correlator circuit. Because the phase shifter and mixer were integrated in the orthogonal demodulator, the error caused by different insertion loss of transmission line could be reduced while complex correlation operation was realized.

In this paper, the signal amplified and filtered by the multistage low-noise amplifier was connected to the power divider, which was divided into two identical half-power signals. A programmable phase shifter was added to one of the input signals to realize the correction of the input phase error. Then, the signal was input to the orthogonal demodulator for complex correlation operation. Then, a differential amplifier was used to suppress the common-mode signal, and a low-pass filter was used to filter the high-frequency component of the signal, so as to obtain the voltage response signal representing the human body temperature. Finally, the amplitude was collected by ADC analog-to-digital converter.

According to the multi-cascade noise figure formula of the receiver,
(32)$$NF=NF_{1}+\frac{NF_{1}-1}{G_{1}}+\frac{NF_{2}-1}{G_{1}G_{2}}+\ldots ,$$
where $NF_{n}$ is the noise figure of the n-th stage amplifier, and $G_{n}$ is the gain of the n-th stage amplifier. Thus, the noise temperature at the RF front end is essentially equivalent to the noise temperature of the amplifier, indicating the importance of the first element on the receiver link. Therefore, the first-stage low-noise amplifier was considered to be directly connected to the antenna when designing the RF front end. The final design block diagram and prototype of the novel interferometric microwave temperature radiometer are shown in Figure 14 and Figure 15.

In order to verify the feasibility of the novel interferometric microwave radiometer, the relevant characteristics of the novel interferometric microwave radiometer were also simulated. The modeling and simulation verification of the novel interferometer analog complex correlator was carried out through ADS, as shown in Figure 16.

In the verification process, the vector network analyzer was used to measure the S parameters of each module and connection line in the novel interferometric analog complex correlator. The corresponding SNP file was derived, and the novel interferometric analog complex correlator was simulated in ADS. In the simulation process, the harmonic simulation control was used to scan the phase of the novel interferometric analog complex correlator to verify its correlation characteristics. The simulation result is shown in Figure 17.

It can be seen from the scanning results that the output complex correlation circle of the novel interference structure had no obvious deformation and center shift; hence, the performance index of the novel interference structure could meet the design requirements.

## 4. Performance Tests of Two Microwave Temperature-Measuring Radiometer Systems

For comparing the performance of two kinds of microwave radiometers, the sensitivity of temperature measurement and the detection of the two systems were further tested.

### 4.1. Temperature Sensitivity of Microwave Temperature Radiometer

In order to ensure the accuracy of the temperature sensitivity tests of the microwave radiometer, the rectangular waveguide antenna was sealed in an aluminum barrel. The antenna was directly facing the water for measurement in the external environment, and the antenna port was surrounded by absorbing medium [3,32]. The aluminum barrel had the function of shielding the electromagnetic interference material to reduce the influence of electromagnetic interference in the aluminum barrel on the measurement results. Then, a water thermometer with a precision of ±0.1 °C/°F was inserted into the aluminum barrel to measure the actual water temperature in real time.

Since the detection characteristics of the microwave radiometer were similar in the four operating bands, only the 12–16 GHz operating band was tested here. The corresponding multichannel switch was adjusted to the corresponding frequency band, and then the output voltage changes of the two microwave temperature radiometers were recorded in real time by controlling the water temperature in the bucket. The measured results of the novel interferometric microwave temperature-measuring radiometer are shown in Figure 18a, and the measured results of the full-power microwave temperature-measuring radiometer are shown in Figure 18b.

Two kinds of radiometer systems were used to measure the water temperature. The sensitivity of the full-power microwave temperature radiometer reached 0.653 K/mV. The sensitivity of the novel interferometric microwave temperature radiometer could reach 0.047 K/mV, which was 13.89-fold higher than that of the full-power radiometer. Therefore, the novel interferometric microwave temperature radiometer could capture smaller temperature changes and had higher measurement sensitivity.

### 4.2. Detection Sensitivity of Microwave Temperature Radiometer

In order to determine the detection sensitivity of both kinds of microwave temperature-measuring radiometers, the input end of the radiometer was added to a high-frequency microwave signal source that could generate any frequency within 0.01–15 GHz through the addition of 60 dB attenuator, to simulate the output signal of the human body in four working frequency bands. Through power scanning of the signal source of the microwave radiometer, we could see the change in output voltage values when the input power of the microwave radiometer changed. These results indicated the detection sensitivity of the microwave radiometer. A larger detection sensitivity denotes a better sensitivity performance of the microwave radiometer.

In this paper, two kinds of microwave temperature-measuring radiometers were used to carry out power scanning on their input terminals, and their output voltage changes were recorded when the input power changed. Figure 19a shows the measured results of the detection sensitivity of the full-power microwave temperature radiometer from 6 GHz to 14 GHz, and Figure 19b shows the measured results of the detection sensitivity of the novel interferometric microwave temperature radiometer from 12 GHz to 14 GHz.

In the actual measurement process, the ratio of the output voltage and input power of the full-power microwave temperature radiometer was linear. The maximum detection sensitivity was 29 mV/dBm at 12 GHz. The ratio of the output voltage to input power of the novel interferometric microwave temperature radiometer was larger than that of the full-power microwave temperature radiometer. When the detection sensitivity was evaluated at 13 GHz, the maximum detection sensitivity was 215 mV/dBm, which was 7.454-fold higher than that of the full-power type. Therefore, the novel interferometric microwave temperature radiometer had better detection characteristics.

## 5. Conclusions

The multiband noncontact temperature measurement microwave radiometer system designed in this paper can penetrate clothing or human tissue to passively receive microwave signals from the internal radiation of the human body. It can provide a technique for noncontact detection, treatment, and monitoring of severe diseases. The innovation and conclusions of this paper are described below.

In this paper, a multiband noncontact microwave radiometer system covering the core temperature field of human body based on the tissue depth model and the relationship between skin depth and electromagnetic frequency was proposed. According to the relationship between human skin depth and electromagnetic wave frequency, four frequency bands of 4–6 GHz, 8–12 GHz, 12–16 GHz and 14–18 GHz were selected for design. The ratio of the half-power beam width to antenna directivity was taken as the constraint condition. By sacrificing the H-plane beam angle appropriately, small and highly directional angular horn antennas of four frequency bands were designed and manufactured by simulation. Each antenna had a frequency range of 4 GHz.

Through the simulation of a full-power microwave radiometer and an analysis of the maladjustment in the additive analog complex correlator, it was found that the nonideality of the device in an additive analog complex correlator would lead to distortion of the output complex correlation circle. On the basis of an analysis of the relationship between temperature measurement error and the uncertainty of the full-power microwave radiometer, a novel hardware structure of an interferometric microwave radiometer was proposed. The experimental results showed that the maximum detection sensitivity of the novel interferometric microwave temperature radiometer was 215 mV/dBm and the temperature sensitivity was 0.047 K/mV. Compared with the full-power microwave temperature radiometer, the detection sensitivity was increased 7.45-fold, and the temperature sensitivity was increased 13.89-fold.

Nevertheless, the design needs to be improved in the future. Firstly, it needs to be adapted to more application scenarios. Secondly, the novel interferometric microwave radiometer prototype needs to be integrated so as to realize a miniaturized handheld multiband noncontact microwave radiometer system. Thirdly, the performance of all bands of the human core temperature field were not tested, and all frequency band indices of the full-power radiometer and the novel interferometric radiometer should be compared experimentally. Lastly, as the environment also affects the measurement results, the novel interferometric microwave temperature radiometer prototype should be implemented in controlled temperature conditions to shield the whole machine, thereby reducing the uncertainty of the temperature measurement and improving its sensitivity.

## Figures and Tables

**Figure 1 micromachines-12-01202-f001:**
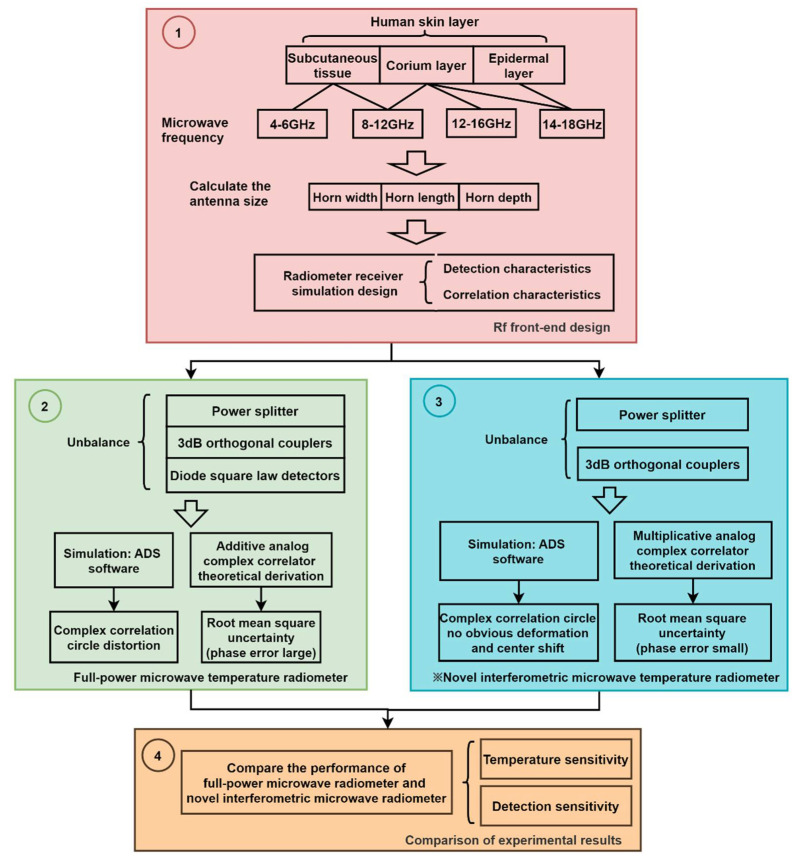
Overall scheme of the technical route.

**Figure 2 micromachines-12-01202-f002:**
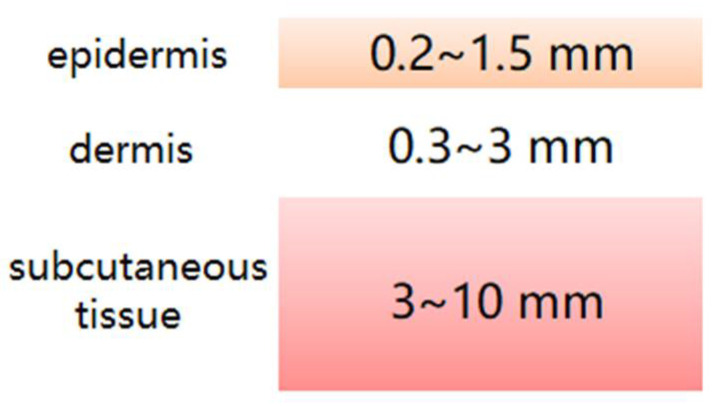
Human tissue depth model.

**Figure 3 micromachines-12-01202-f003:**
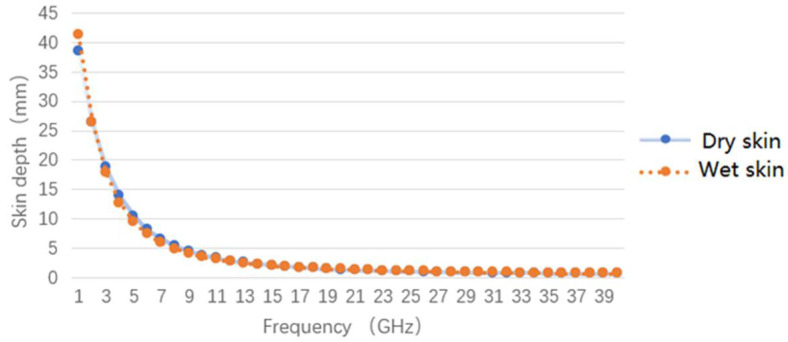
Relationship between skin depth and microwave frequency.

**Figure 4 micromachines-12-01202-f004:**
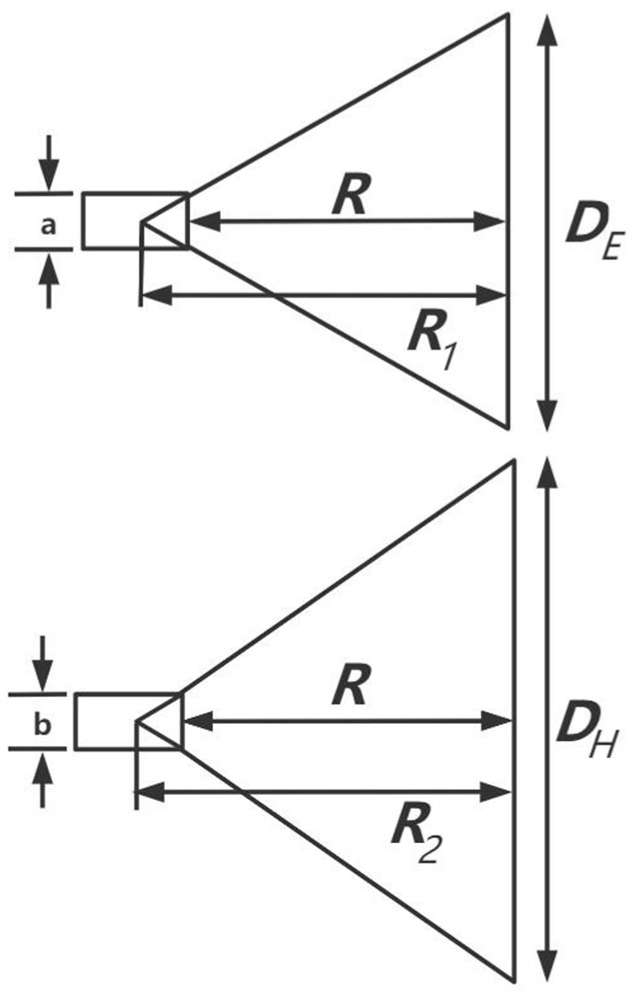
Antenna schematic diagram of E-plane and H-plane.

**Figure 5 micromachines-12-01202-f005:**
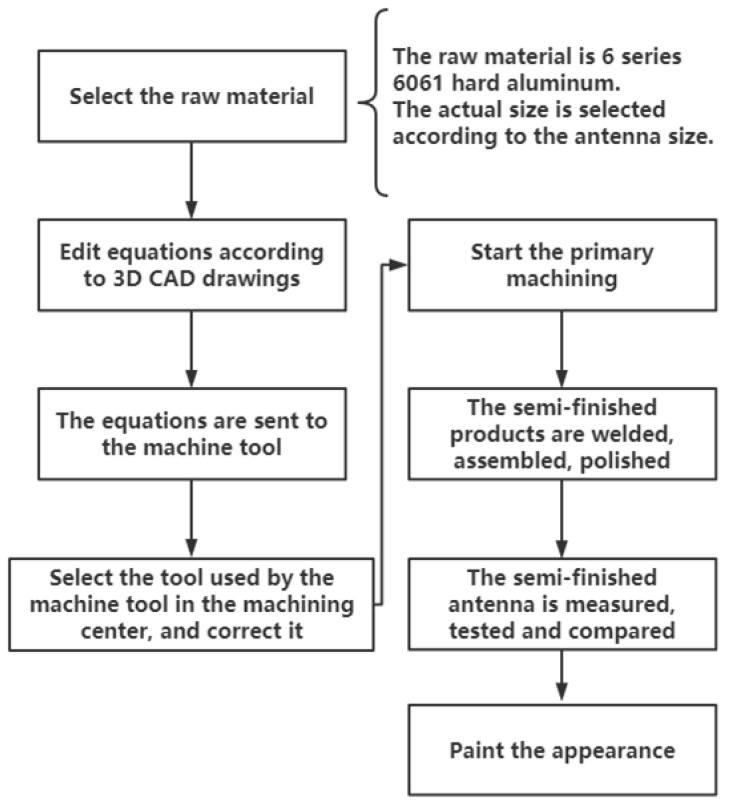
Antenna processing flow chart.

**Figure 6 micromachines-12-01202-f006:**
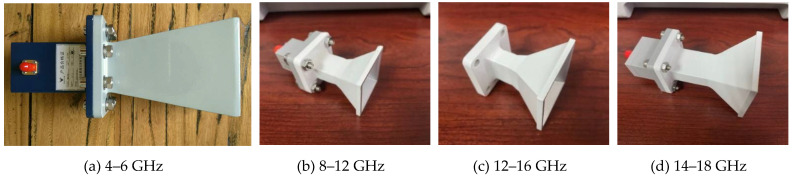
Images of each frequency cone horn antenna.

**Figure 7 micromachines-12-01202-f007:**
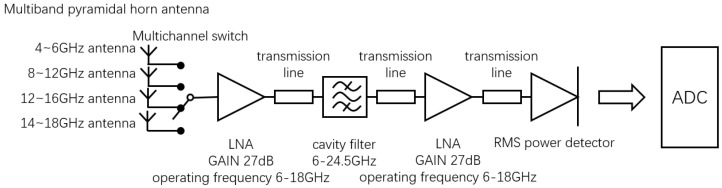
Block diagram of multiband full-power microwave temperature radiometer.

**Figure 8 micromachines-12-01202-f008:**
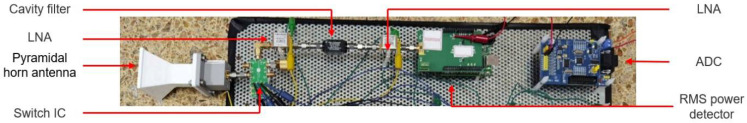
Full-power microwave temperature radiometer prototype.

**Figure 9 micromachines-12-01202-f009:**
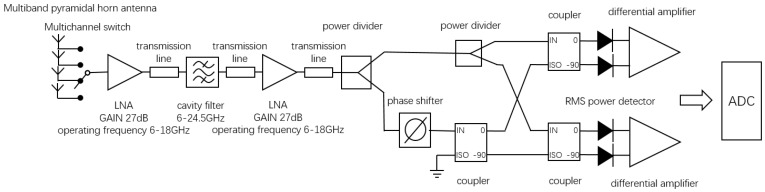
Block diagram of full-power microwave radiometer characteristics.

**Figure 10 micromachines-12-01202-f010:**
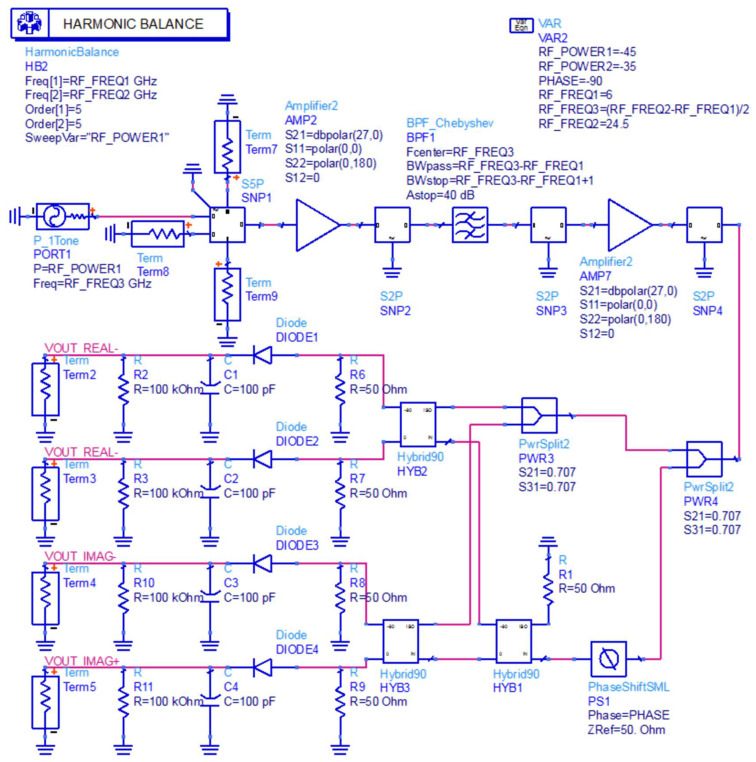
Schematic diagram of full-power microwave radiometer simulation.

**Figure 11 micromachines-12-01202-f011:**
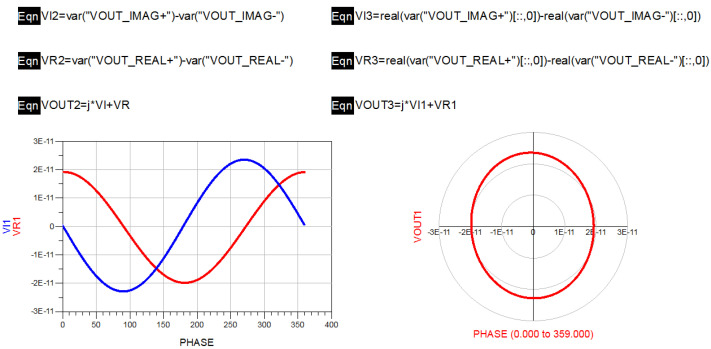
Phase scanning results of the full-power microwave temperature radiometer.

**Figure 12 micromachines-12-01202-f012:**
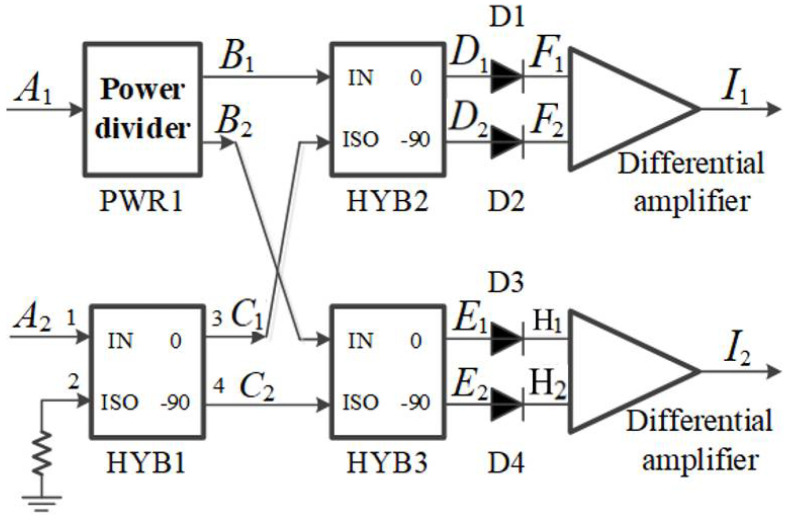
Schematic block diagram of an additive analog complex correlator.

**Figure 13 micromachines-12-01202-f013:**
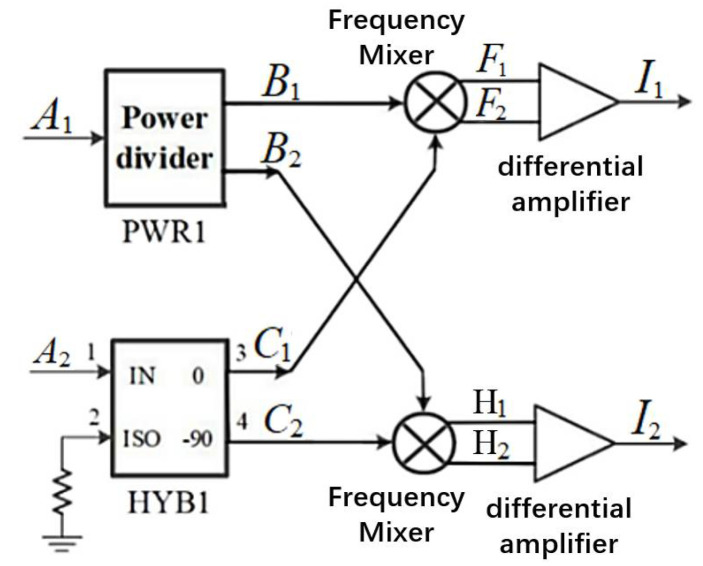
Block diagram of a multiplicative analog complex correlator.

**Figure 14 micromachines-12-01202-f014:**
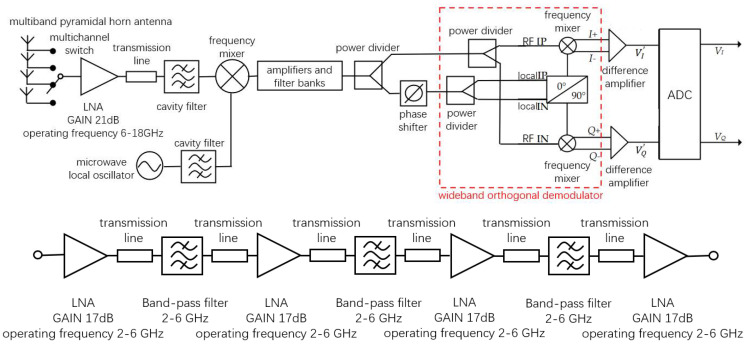
Design block diagram of novel interferometric microwave temperature radiometer. Upper panel: overall block diagram of the novel interferometric microwave radiometer; lower panel: partial block diagrams of amplifiers and filter banks.

**Figure 15 micromachines-12-01202-f015:**
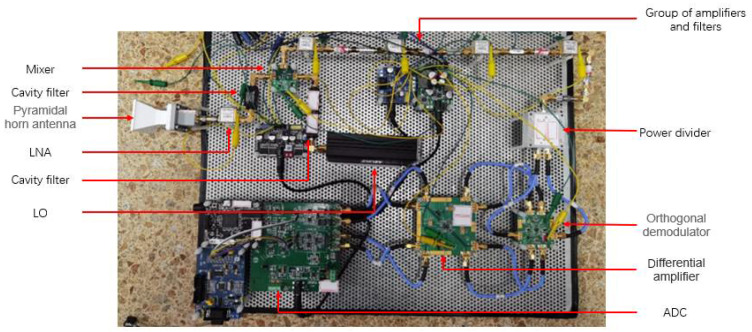
Prototype of novel interferometric microwave temperature-measuring radiometer.

**Figure 16 micromachines-12-01202-f016:**
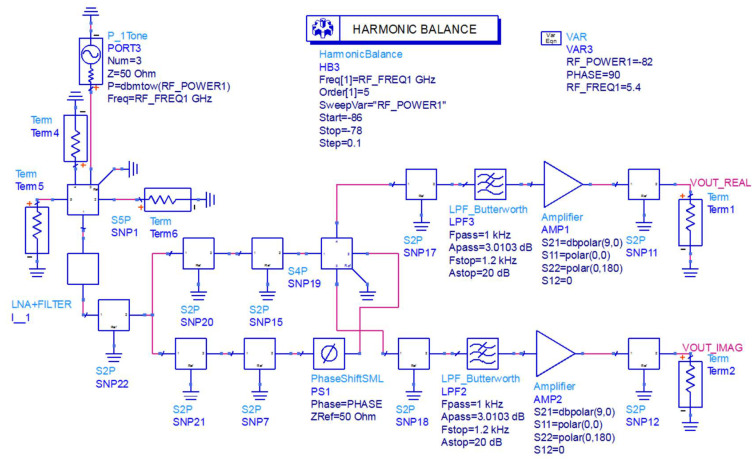
Simulation schematic diagram of the novel interferometric analog complex correlator.

**Figure 17 micromachines-12-01202-f017:**
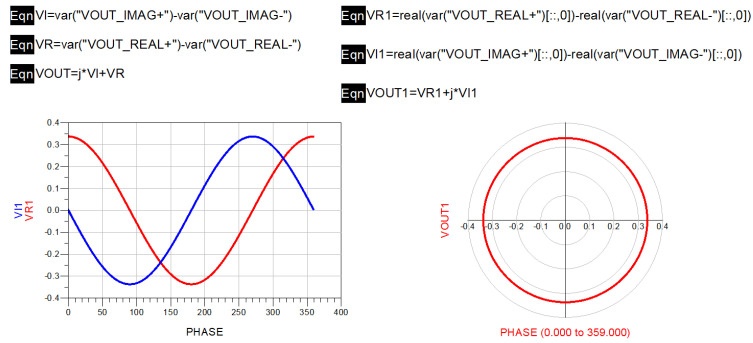
Phase scanning results of novel interferometric analog complex correlator.

**Figure 18 micromachines-12-01202-f018:**
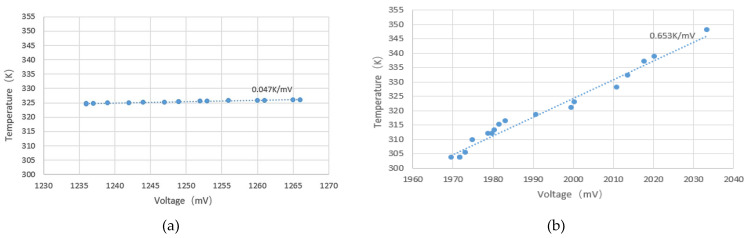
(**a**) Measurement results of the sensitivity of the novel interferometric microwave temperature radiometer; (**b**) measured results of the temperature sensitivity of the full-power microwave temperature radiometer.

**Figure 19 micromachines-12-01202-f019:**
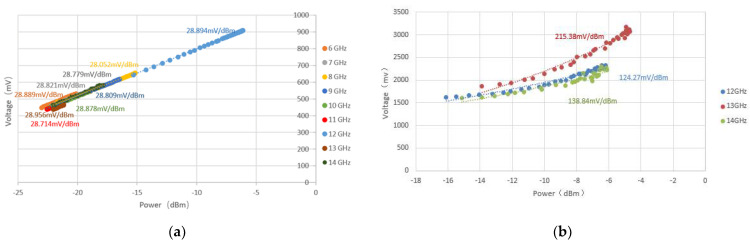
(**a**) Measured results of the detection sensitivity of full-power microwave temperature radiometer; (**b**) measured results of the detection sensitivity of the novel interferometric microwave temperature radiometer.

**Table 1 micromachines-12-01202-t001:** The relationship between the frequency bands of the multiband noncontact thermometer and skin depth and human tissue.

Microwave Frequency	4–6 GHz	8–12 GHz	12–16 GHz	14–18 GHz
Skin depth of dry skin (mm)	8.2–13.9	2.9–5.3	1.9–2.9	1.5–23
Skin depth of wet skin (mm)	7.4–12.7	2.7–4.9	1.8–2.7	1.5–2.2
Human tissue	Subcutaneous tissue	Corium layer and subcutaneous tissue	Corium layer	Epidermal layer and corium layer

**Table 2 micromachines-12-01202-t002:** The size of the corresponding pyramid horn antenna designed. *L* is the length of the antenna.

Antenna Frequency	$D_{E}$	$D_{H}$	L
4–6 GHz	88.04 mm	89.05 mm	49.08 mm
8–12 GHz	46.63 mm	46.23 mm	29.01 mm
12–16 GHz	34.47 mm	34.18 mm	21.14 mm
14–18 GHz	30.16 mm	29.91 mm	17.70 mm

**Table 3 micromachines-12-01202-t003:** Simulation results of pyramid horn antenna index.

Antenna Frequency	VSWR	Far-Field Radiation Model	Pre/Post Suppression Ratio	Far-Field Gain	3 dB Beam Angle
4–6 GHz	≤1.98	38.1 V/m	18.65	15.88 dB	38.5°
8–12 GHz	≤1.48	34.9 V/m	17.46	14.90 dB	29.4°
12–16 GHz	≤1.44	34.7 V/m	16.44	14.36 dB	30.2°
14–18 GHz	≤1.41	35.0 V/m	19.68	14.45 dB	30.9°

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
