# Peer review of "Design and Implementation of Multiband Noncontact Temperature-Measuring Microwave Radiometer"

_micromachines, 2021, doi:10.3390/mi12101202_

Round 1

Reviewer 1 Report

See comments in the attached file.

Author Response

Dear Reviewer 1,‎

Thank you for your report. ‎

We would also like to thank you, for taking the time to read our paper and for your important comments and remarks. Please see the attachment.

Please see the attachment for the detailed reply to your comments.

Yours sincerely

‎Jingyan Ma

Reviewer 2 Report

Dear authors,

I've fonund your paper very interesting and I thank you for a work well done. I have the following comments and suggestions for you.

1) I have learned that there should be a space between numbers and units, i.e. 14 GHz and not 14GHz. Maybe I'm wrong, but please check.

Line 44: you wrote "Because the clothing has a very weak attenuation effect on microwave", I propose ""...on microwaves"  or "...on microwave radiation" or "...on microwave signals"

Line 53: stat the phrase with "A..."

Line 72: change the "but" to "However"

Line 128: change to "...usually the abdomen, buttocks and thighs..."

Figure 2 : different fonts are used for the numbers

Line 190: you wrote "Because the  literature  based  on  microwave  ...", I propose "The  literature  on  the subject of microwave ..."

Figure 8 and Figure 14: These are ADS schematic diagrams, some editors do not agree with that, I'm not sure about MDPIs policy on this matter, however to me it gives useful information

Line 347: changer "Where" to "where" and skip the tab

Figure 16: I would interchange the axes to have mV on y-axis and T on x-axis

Figure 17: The input power is before or after taking the 60 dB attenuator into account?

The frequency range tested is not the same, can really a comparison be done correctly?

Line 426: I admit that the curve is no longer linear, but is it really exponential? Maybe a zoom in on the last part of the 13GHz curve would be informative?

Some references (8, 10) includes "etc." insted of "et.al." this is a novelty for me, is it correct to do that?

Some references, (13, 16, 21, 23...) seem to point towards internal university documents or documents are badly referenced, can't you find references that are available to all of us?

Thank you and have a nice day

Author Response

Dear Reviewer 2,‎

Thank you for your report. ‎

We would also like to thank you, for taking the time to read our paper and for your important comments and remarks.

Please see the attachment for the detailed reply to your comments.  

Yours sincerely,

‎Jingyan Ma

Round 2
